# Biological Effects of Bioresorbable Materials in Alveolar Ridge Augmentation: Comparison of Early and Slow Resorbing Osteosynthesis Materials

**DOI:** 10.3390/ma14123286

**Published:** 2021-06-14

**Authors:** Hotaka Kawai, Shintaro Sukegawa, Keisuke Nakano, Kiyofumi Takabatake, Sawako Ono, Hitoshi Nagatsuka, Yoshihiko Furuki

**Affiliations:** 1Department of Oral Pathology and Medicine, Graduate School of Medicine, Dentistry and Pharmaceutical Sciences, Okayama University, Okayama 700-8525, Japan; de18018@s.okayama-u.ac.jp (H.K.); pir19btp@okayama-u.ac.jp (K.N.); gmd422094@s.okayama-u.ac.jp (K.T.); jin@okayama-u.ac.jp (H.N.); 2Department of Oral and Maxillofacial Surgery, Kagawa Prefectural Central Hospital, Takamatsu 760-8557, Japan; furukiy@ma.pikara.ne.jp; 3Department of Pathology, Kagawa Prefectural Central Hospital, Takamatsu 760-8557, Japan; de19008@s.okayama-u.ac.jp

**Keywords:** poly L-lactic acid, uncalcined and unsintered hydroxyapatite, polyglycolic acid, alveolar ridge augmentation

## Abstract

The purpose of this study was to investigate the bone healing properties and histological environment of a u-HA/PLLA/PGA (u-HA—uncalcined and unsintered hydroxyapatite, PLLA—Poly L-lactic acid, PGA—polyglycolic acid) composite device in humans, and to understand the histological dynamics of using this device for maxillofacial treatments. Twenty-one subjects underwent pre-implant maxillary alveolar ridge augmentation with mandibular cortical bone blocks using u-HA/PLLA or u-HA/PLLA/PGA screws for fixation. Six months later, specimens of these screws and their adjacent tissue were retrieved. A histological and immunohistochemical evaluation of these samples was performed using collagen 1a, ALP (alkaline phosphatase), and osteocalcin. We observed that alveolar bone augmentation was successful for all of the subjects. Upon histological evaluation, the u-HA/PLLA screws had merged with the bone components, and the bone was directly connected to the biomaterial. In contrast, direct bone connection was not observed for the u-HA/PLLA/PGA screw. Immunohistological findings showed that in the u-HA/PLLA group, collagen 1a was positive for fibers that penetrated vertically into the bone. Alkaline phosphatase was positive only in the u-HA/PLLA stroma, and the stroma was negative for osteocalcin. In this study, u-HA/PLLA showed a greater bioactive bone conductivity than u-HA/PLLA/PGA and a higher biocompatibility for direct bone attachment. Furthermore, u-HA/PLLA was shown to have the potential for bone formation in the stroma.

## 1. Introduction

Resorbable osteosynthesis materials have become widespread in recent years. Titanium osteosynthesis materials may cause complications over time, and may need to be removed, as seen after the treatment of fractures and jaw deformities in the maxillofacial region [1,2]. On the other hand, absorbable materials do not need to be removed because they are absorbed and decomposed in the human body. Therefore, these materials simplify the postoperative management of the patients. In addition, the use of these materials is not only minimally invasive, requiring no re-surgery, but is also useful for the healthcare economy.

Poly L-lactic acid (PLLA) has been used as a bioabsorbable material that is easy to process, owing to its high mechanical properties and ease of processing. However, PLLA is not only difficult to incorporate into the body because of its hydrophobicity, but also has a slow decomposition rate owing to its difficult-to-react material, which may cause an inflammatory reaction after implantation into the body [3]. Therefore, attempts have been made to add various materials to PLLA to increase its rate of degradation and to enhance the intended biomedical effects.

The PLLA and polyglycolic acid (PLLA/PGA) copolymer has the advantage of rapid decomposition within one year after implantation in the body [3,4]. The material containing PGA has a better absorption and decomposition rate than PLLA alone. However, its mechanical strength is lower than that of PLLA alone. In addition, both PLLA/PGA and PLLA alone, as resorbable osteosynthesis materials, do not have biological activity characteristics such as bone conduction and osteosynthesis ability. In response, bioabsorbable fixation devices made of high-strength uncalcined and unsintered hydroxyapatite (u-HA) and PLLA composites have been developed to solve the mechanical and biological problems of life-long resorbable osteosynthesis materials [5,6]. Composed of u-HA and PLLA, this bioabsorbable device is manufactured by a compression molding strengthening process and a forging process that incorporates machining. Owing to its composition and special manufacturing process, this device has achieved a higher mechanical strength and biological activity [7,8,9,10]. These advantages have led to the use of bioactive screws alone in the maxillofacial region [11].

Thus, although the u-HA/PLLA material has superior biological properties, the drawback of slow degradation was not remedied. The slow degradation of absorbable materials has been reported to induce persistent patient discomfort [12].

Recently, new absorbable materials, unfired/unsintered hydroxyapatite and poly-L-lactide-co-glycolide (u-HA/PLLA/PGA), were developed to address this issue, while also preserving the advantages of the existing materials. According to its developers, u-HA/PLLA/PGA has a resorption period of approximately 2–3 years, which is superior to the five years taken by u-HA/PLLA to resorb completely. Ngo et al. [13]. reported excellent bone formation using u-HA/PLLA/PGA in jaw bone defects in animal studies. However, the biocompatibility and bioactive bone conduction properties of u-HA/PLLA/PGA in humans remain unclear.

Therefore, it is necessary to clinically and histologically evaluate the bone regeneration ability of u-HA/PLLA/PGA in the human body. The purpose of the present study was to investigate the bone healing properties and histological environment of a u-HA/PLLA/PGA composite device in humans, and to understand the histological dynamics of using this device in maxillofacial treatments.

## 2. Materials and Methods

### 2.1. Preparation of u-HA/PLLA Composite Screws

In this study, forged composite screws of u-HA/PLLA (Super FIXSORB MX^®^; Teijin Medical Technologies Co., Ltd., Osaka, Japan) and u-HA/PLLA/PGA (Super FIXSORB EX^®^; Teijin Medical Technologies Co., Ltd., Osaka, Japan) were used.

The following characteristics were common to both screws: diameter 2.0 mm, u-HA particle size 0.2–20 μm (average size, 3–5 μm), Ca/P 1.69 (mol, ratio), and CO_3_^2−^ 3.8 (mol%).

On the other hand, the length (u-HA/PLLA 8–12 mm and u-HA/PLLA/PGA 7 mm) and mixing ratio (u-HA/PLLA 30/70 and u-HA/PLLA + PGA 10/90) of the screws differed from each other.

### 2.2. Subjects

This study was approved by the Ethics Committee of the Kagawa Prefectural Central Hospital (approval number 1021, Takamatsu, Japan). In this study, 21 patients (13 men and 8 women; mean age 47.7 ± 21.4 years old, range 16–74 years old) with a residual ridge width of <4 mm, needing maxillary alveolar ridge augmentation prior to implant placement, were included after obtaining their consent (Table 1). An oral and maxillofacial surgeon (Sukegawa) performed the surgery for all patients between April 2019 and May 2020 at the Department of Oral and Maxillofacial Surgery, Kagawa Prefectural Central Hospital (Takamatsu, Japan).

u-HA/PLLA screws were selected for the procedures performed until February 2020 and u-HA/PLLA/PGA screws for those performed from March 2020 until completion of the study period.

### 2.3. Surgical Alveolar Bone Augmentation Procedure 

During the surgical procedure, a cortical bone block of a volume necessary for alveolar bone augmentation in the anterior maxilla was collected from the buccal cortical bone of the mandibular ramus. It was then fixed to the recipient site using u-HA/PLLA or u-H/PLLA/PGA screws. The screw fixing method involved (1) the drilling of bone to form a hole, (2) screw tapping, and (3) insertion of screws into the holes formed by self-tapping. The screw insertion torque was 5 N. The number of screws used was that required to ensure the stable fixation of the block bone. The block bone and recipient site were contoured to improve the adaptation of the graft to the recipient bed. No bone graft was placed in the gap between the recipient site and the bone.

### 2.4. Sample Collection

After allowing approximately 6 months for bone healing, we planned the dental implant placement surgery using computed tomography. At the same time, the position of the u-HA/PLLA and u-HA/PLLA/PGA screws was confirmed. Instead of cavity formation for implant placement with drilling, specimens were collected using a 2.0 mm diameter trephine bur (ACE Surgical Supply Company, Inc., Brockton, MA, USA). Absorbable screws placed outside the implant placement position were not sampled and were excluded from this study. All of the surgeries were performed by an oral and maxillofacial surgeon (Sukegawa) at a single facility.

### 2.5. Histological Evaluation

The collected samples were fixed in 4% paraformaldehyde for 12 h and decalcified in 10% ethylenediaminetetraacetic acid (EDTA) at 4 °C for 14 days. The samples were then sequentially dehydrated in 70% ethanol and were embedded in paraffin. Serial sections (5-μm thickness) were prepared. Sections were stained with hematoxylin and eosin (HE) or Masson’s trichrome staining (40251, MUTO pure chemicals, Tokyo, Japan) to observe collagen deposition.

### 2.6. Immunohistochemistry

Immunohistochemistry (IHC) was performed using the antibodies listed in Table 2. Following antigen retrieval, sections were treated with 10% normal serum for 15 min and then incubated with primary antibodies at 4 °C overnight. The signals were enhanced using the avidin-biotin complex method (Vector Lab, Burlingame, CA, USA). Color development was performed with 3, 3′-diaminobenzidine (Histofine diaminobenzidine substrate kit; Nichirei, Tokyo, Japan), and the staining results were observed with an optical microscope.

To investigate the relationship between the bone and fibrous stroma, we performed IHC staining of the samples from both groups. First, we stained collagen 1a, a marker of collagen fibers. Next, to check the bone-formation potential of the stroma, alkaline phosphatase (ALP) staining was performed.

## 3. Results

### 3.1. Clinical Evaluation

After 6 months of bone healing following anterior maxillary alveolar bone augmentation, all patients had a sufficient bone width for implant placement. The transplanted cortical bone block was fully engrafted in all patients (Figure 1). No complications were observed after dental implant placement, and all cases showed satisfactory results with the final prosthesis. Eight specimens with u-HA/PLLA screws were obtained from the augmented area using a trephine bur at the time of implant placement. Among them, five specimens in which the position of the implant and u-HA/PLLA screw placements matched, were examined histologically. In addition, very interestingly, the u-HA/PLLA screw was not altered, but the u-HA/PLLA/PGA screw was fragile.

### 3.2. Histopathological Evaluations

Twelve u-HA/PLLA and nine u-HA/PLLA/PGA specimens were examined using hematoxylin and eosin (HE) staining. In eight (66.6%) of the u-HA/PLLA specimens and nine (33.3%) of the u-HA/PLLA/PGA specimens, we observed both screw and surrounding bone. Therefore, there was a difference in the proportion of specimens demonstrating both screw and surrounding bone in the two groups.

HE staining was performed to evaluate the histological differences between both groups. Fibrous tissue was seen surrounding the screw in the u-HA/PLLA group. The histological features of the fibrous tissue were uniform across all of the specimens of this group, and there were no signs of inflammation or bleeding (Figure 2). Foreign body giant cells were not present in the stromal tissue (Figure 2g,h). Furthermore, this fibrous tissue contained bone tissue and was continuous around the screw (Figure 2c,d,g,h). The u-HA/PLLA/PGA groups also revealed fibrous tissue surrounding the screw (Figure 3). The fibrous tissue was dense, and many fibroblasts were observed. This fibrous tissue surrounded the bone tissue. There were no signs of inflammation, and foreign body giant cells (Figure 3b,c,f,g) were not observed. These findings indicate that both u-HA/PLLA and u-HA/PLLA/PGA groups are highly biocompatible. However, the relationship between the bone and stromal fibrous tissue was different in the two groups.

### 3.3. Fibrous Tissue Evaluation

To reveal the differences in the fibrous tissue characteristics between the groups, we performed Masson’s trichrome staining (Figure 4). In the u-HA/PLLA group, blue-stained collagen fibers were observed in the stromal area (Figure 4a). The fibers were present as bundles perpendicular to the bone, penetrating it (Figure 4b,c). In addition, blue-stained bands were observed in the bundled bones, and the blue band contained cells such as bone cells (Figure 4b,c). On the other hand, in the u-HA/PLLA/PGA group, the blue-stained collagen fibers ran parallel to the bone (Figure 4d). The fibers were thin and did not form bundles. No blue band was observed around the bone, unlike the u-HA/PLLA group (Figure 4e,f). These results indicated that although both materials were associated with fibrosis, the fibrosis characteristics differed in both groups. In particular, in the u-HA/PLLA group, the collagen fibers were directly connected to the bone tissue. These findings suggest that in the u-HA/PLLA group, the stromal fibers were involved in new bone formation.

### 3.4. Immunohistochemical Evaluations

Collagen 1a was positive in the fibers that penetrated vertically into the bone in the u-HA/PLLA group (Figure 5a,d). ALP staining was positive only in the u-HA/PLLA group (Figure 5b). It was absent in the stroma in the u-HA/PLLA/PGA group (Figure 5e). Osteocalcin, an osteoblast marker, was also stained. Osteocalcin positive cells were observed at the bone surface or bone-included cells and stromal round cells in the u-HA/PLLA group (Figure 5c). In the u-HA/PLLA/PGA group, the stroma was negative for osteocalcin (Figure 5f). These findings indicated that the u-HA/PLLA stroma has the potential for bone formation.

## 4. Discussion

In this study, reliable pre-implant bone augmentation using a block bone was observed with both u-HA/PLLA and u-HA/PLLA/PGA screws. The u-HA/PLLA materials were highly biocompatible, especially those that directly bonded to the alveolar bone. It was suggested that u-HA/PLLA, which has a high concentration of u-HA, has bone conductivity.

It is very important to histologically prove the difference in the outcomes of the u-HA/PLLA and u-HA/PLLA/PGA materials used in this study. In general, synthetic polymer materials can cause a decrease in strength, but this material containing u-HA guarantees sufficient strength when undergoing a special compression molding strengthening process [6,14]. u-HA/PLLA materials have advantages such as a high biological activity, osteoconductivity, biocompatibility, and stable complete degradation [8,12,15,16,17,18], owing to the presence of uncalcined hydroxyapatite compounds [19]. Furthermore, we observed that u-HA/PLLA had a direct bond with the bone in this study. When cells touch the surface of a material, they usually adhere and spread. The first stage of this cell/material interaction depends on the characteristics of the material surface in order to determine the behavior of the cell upon contact with the material. Osteoblasts have been found to preferentially attach to HA particles via filopodia, demonstrating that HA provides a favorable anchoring site for human osteoblast adhesion [20].

In contrast, no direct binding to bone was observed with u-HA/PLLA/PGA. There was also a significant difference when preparing the specimens collected with a trephine bur. While the u-HA/PLLA samples were easy to prepare, most u-HA/PLLA/PGA samples were difficult to prepare because the bone and screw were easily peeled off. The difference between u-HA/PLLA and u-HA/PLLA/PGA is the difference in the u-HA concentration. The u-HA content is 30% in u-HA/PLLA and 10% in u-HA/PLLA/PGA. The structural differences are clearly visible with an electron microscope. The direct binding of the material to bone is thought to be due to the difference in the u-HA concentration.

u-HA/PLLA materials have been reported to induce bone formation [8]. In this study, u-HA/PLLA was in direct contact with the bone, and new bone formation was observed around the materials. In addition, there were very few inflammatory findings, and no foreign body giant cells were observed. These results were consistent with the u-HA-PLLA histological findings reported previously [8,17]. The new finding in this study is that collagen fibers present in the stroma of u-HA/PLLA are involved in osteogenesis. In the stroma of the u-HA/PLLA group, collagen fibers penetrated perpendicular to the new bone, and the fibers were taken up into the new bone. These findings are similar to those of the bundle bone structure [21]. Bundle bone is a new bone that forms on the surface of the alveolar bone, found in the periodontal ligament tissue. Embedded within this bone are the extrinsic collagen fiber bundles of the periodontal ligament that are mineralized only at their periphery. Bundle bone thus provides attachment to the periodontal ligament fiber bundles that are inserted into it. Histologically, bundle bone generally contains less intrinsic collagen fibrils than lamellar bone, and exhibits a coarse-fibered texture. Because this experiment was performed on the tissue around the screw inserted into the mandible, the results of the u-HA/PLLA group were considered to be a phenomenon similar to that occurring in the alveolar bone. This material will be useful in aesthetic areas, such as the maxillary anterior tooth region, which requires a reliable amount of bone formation for prosthetic rehabilitation [22].

The bone is composed of a bone matrix and bone cells. The main components of the bone matrix are collagen fibers and apatite, the ratio of which is 35% collagen fibers and 65% apatite. Collagen 1a is a marker for collagen fibers [23]. In the u-HA/PLLA group, collagen 1a positive fibers were observed in the bundle bone. In the u-HA/PLLA/PGA group, collagen 1a positive fibers ran parallel to the bone. These findings indicate that stromal fibers are collagen 1a positive collagen fibers.

Our study showed elevated ALP and osteocalcin expression levels in the stroma of u-HA/PLLA. Osteocalcin is a marker of osteoblasts and osteoblastic progenitor cells [24]. Our data indicated that osteoblast progenitor cells were localized to the u-HA/PLLA stromal area, whereupon osteoblast differentiation occurred, as demonstrated by the expression of ALP by spindle-shaped stromal cells. These ALP-positive stromal cells are considered to have bone formation potential [23]. ALP and osteocalcin expression were only observed in the stroma of the u-HA/PLLA group, with no expression in the u-HA/PLLA/PGA stroma. These findings indicate that the u-HA/PLLA stroma induced bone formation.

Since 2004, after Chacon et al. [25] reported block bone grafting with resorbable screws in animal models, a few similar studies have been reported. In 2006, Raghoebar et al. [26] reported a comparative study of biodegradable screws consisting of poly-DL-lactide acid (PDLLA) and titanium screws for the fixation of bone grafts in humans. The study reported no histological signs of a significant inflammatory response to PDLLA materials, except for more abundant giant cells, in comparison with the response to titanium screws. They also showed the presence of a fibrous connective tissue attachment between the PDLLA screws and bone. Quereshy et al. [27] published a study on the use of absorptive screws to fix human block grafts, showing that PGA/PLLA screws do not adversely affect graft integration and viability. Unfortunately, this study did not include histological evaluation. On the other hand, the present study demonstrates the histological evaluation of u-HA/PLLA and u-HA/PLLA/PGA in humans.

Our research has two limitations. The first is that the sample size is small and there is no control group setting. It was difficult to enroll many cases and set up control groups in this human-based study. In the future, we hope to study more patients over an ex-tended research duration. Regarding the second limitation of this study, the tissue changes associated with these materials were not assessed over time and were considered as a short follow-up study. In this study, we evaluated the timing of dental implant placement after bone formation and samples were collected. In particular, u-HA/PLLA took a long time to absorb; therefore, it is desirable to evaluate the changes over time in the future. However, this research method has already been established and evaluated, and a comparison between u-HA/PLLA and u-HA/PLLA/PGA was made on that basis. [8,17] As a result, to the best of our knowledge, the results of this study elucidate the bone healing properties and histological environment of the bone healing of u-HA/PLLA and u-HA/PLLA/PGA bioabsorbable materials in human maxillofacial bone; this is the first study to do so. This is an interesting and important description of the in vivo reaction of u-HA/PLLA and u-HA/PLLA/PGA bioabsorbable materials.

## 5. Conclusions

In this study, reliable pre-implant bone augmentation was performed using u-HA/PLLA and u-HA/PLLA/PGA screws. We observed that u-HA/PLLA directly attached to the alveolar bone, and direct collagen fiber bone formation was confirmed. In addition, the stroma associated with u-HA/PLLA had a bone-forming ability. However, direct collagen fiber bone formation was not confirmed in relation to u-HA/PLLA/PGA. It is suggested that u-HA/PLLA is a more suitable material for aggressive bone formation.

## Figures and Tables

**Figure 1 materials-14-03286-f001:**
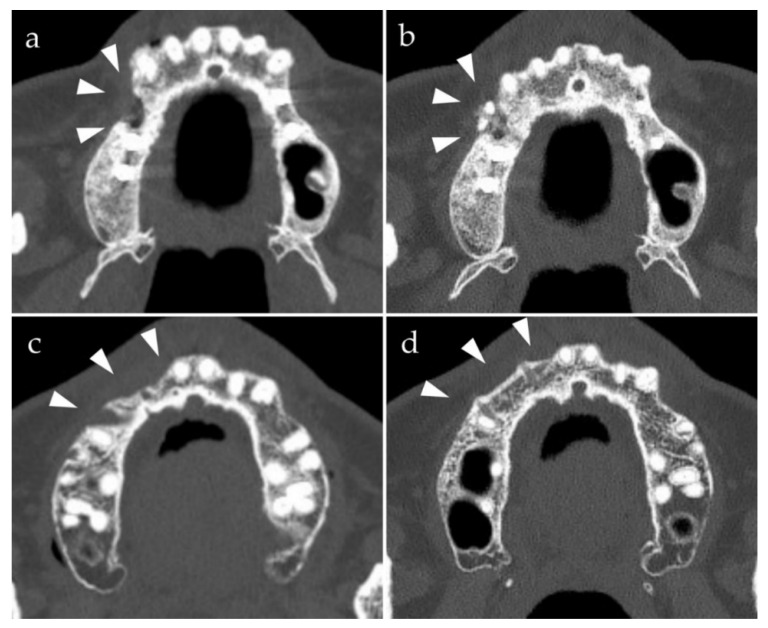
Preoperative and postoperative computed tomography axial images of bone augmentation using u-HA/PLLA and u-HA/PLLA/PGA (arrowheads, bone augmentation site). The density of the u-HA/PLLA screws is greater than that of the u-HA/PLLA/PGA screws. (**a**) Before placement of u-HA/PLLA screws; (**b**) 6 months after the placement of u-HA/PLLA screws; (**c**) before placement of u-HA/PLLA/PGA screws; (**d**) 6 months after placement of u-HA/PLLA/PGA screws.

**Figure 2 materials-14-03286-f002:**
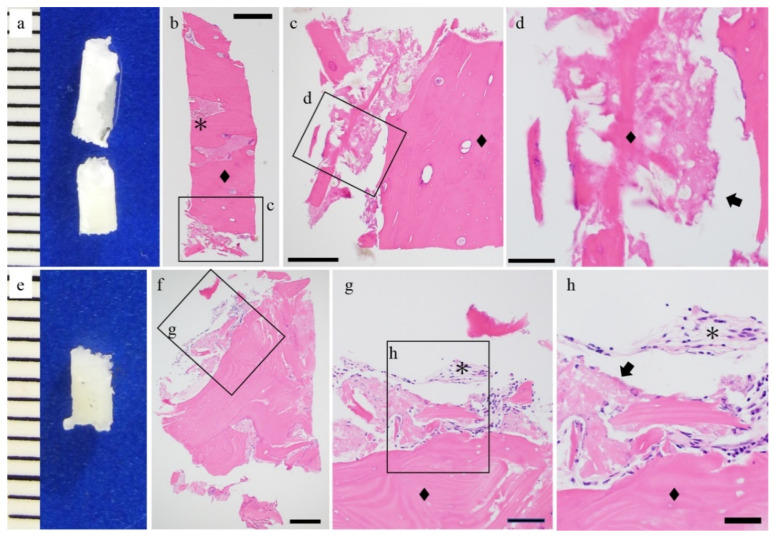
Histological findings of the u-HA/PLLA cases. (**a**,**e**) Macro findings and hematoxylin and eosin staining. (**b**–**d**,**f**–**h**) Hematoxylin and eosin staining. The hydroxyapatite/poly-l-lactide (u-HA/PLLA) screw (arrow) and bone (◆) are in direct contact. Connective tissue was observed (*). Bar: (**b**) 500 μm, (**c**,**f**) 200 μm, (**d**,**g**) 100 μm, and (**h**) 50 μm.

**Figure 3 materials-14-03286-f003:**
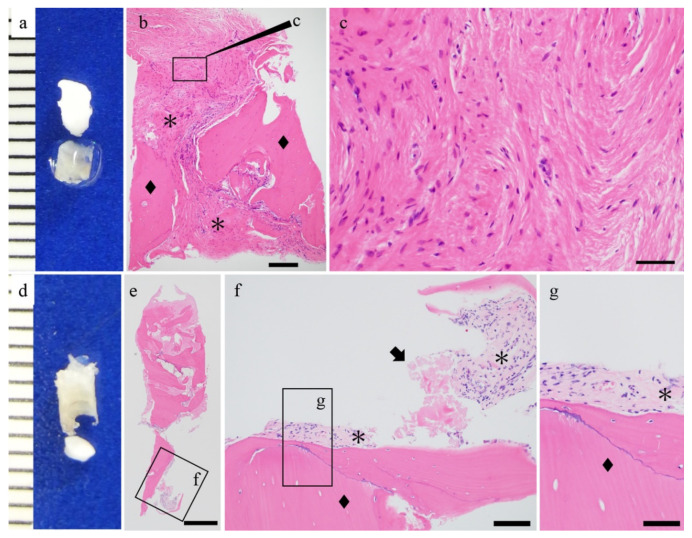
Histological findings of the u-HA/PLLA/PGA cases. (**a**,**d**) Macro findings and hematoxylin and eosin staining. (**b**,**c**,**e**–**g**) Hematoxylin and eosin staining. u-HA/PLLA/PGA screw (arrow) and bone (◆) and connective tissue were observed (*). Bar: (**e**) 500 μm, (**b**) 200 μm, (**f**) 100 μm, and (**c**,**g**) 50 μm.

**Figure 4 materials-14-03286-f004:**
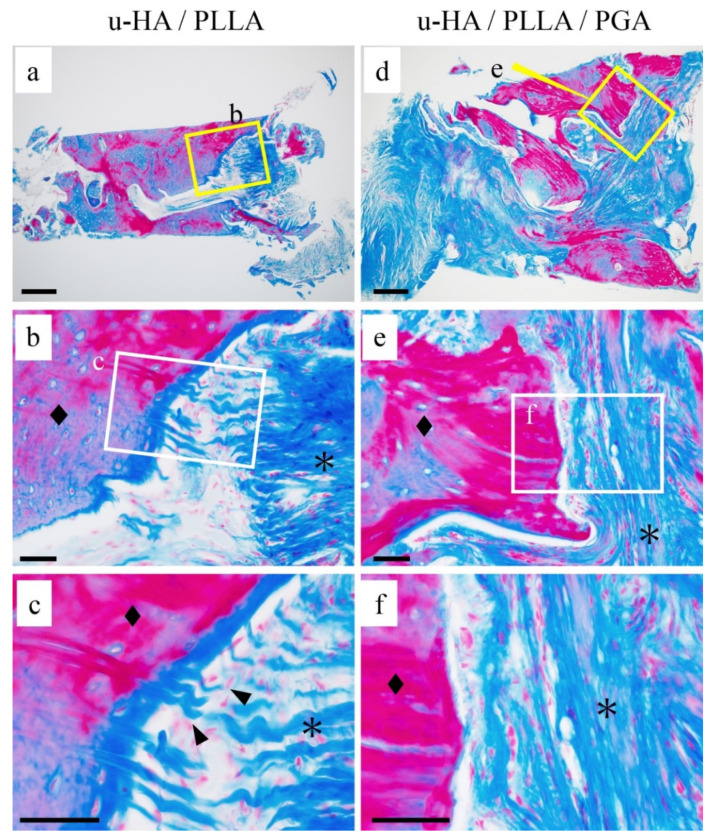
The differences in connective tissue findings. (**a**–**f**) Masson trichrome staining. (**a**–**c**) u-HA/ PLLA group: The fibers were present as bundles penetrating perpendicularly into the bone (c: ▲). (**d**–**f**) u-HA/ PLLA/PGA group: collagen fibers ran parallel to the bone. Bone (◆) and connective tissue (*). Bar: (**a**,**d**) 200 μm and (**b**,**c**,**e**,**f**) 50 μm.

**Figure 5 materials-14-03286-f005:**
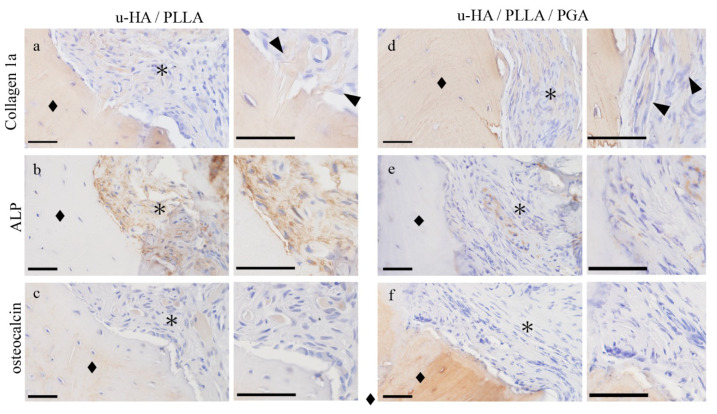
The immunohistochemical characteristics of the connective tissue samples. (**a**,**d**) Collagen 1a, (**b**,**e**) ALP, and (**c**,**f**) osteocalcin staining. (**a**–**c**) u-HA/PLLA group: The fibers are seen as bundles penetrating the bone perpendicular to it. Fibers are positive for collagen 1a ((**a**) ▲). ALP and osteocalcin expression in the connective tissue. (**d**–**f**) u-HA/PLLA/PGA group: Collagen fibers run parallel to the bone ((**d**) ▲). Bone (◆) and connective tissue (*). Bar: 50 μm.

**Table 1 materials-14-03286-t001:** Patient distribution for alveolar bone augmentation in both groups.

Clinical Variable	u-HA/PLLA	u-HA/PLLA/PGA
Age	50.0 ± 20.5	41.9 ± 22.6
Gender		
Male	6	7
Female	3	5
Bone healing period (days)	204.7 ± 13.7	191.2 ± 19.2

**Table 2 materials-14-03286-t002:** Antibodies used for immunohistochemistry.

Antigen Targeted by Primary Antibody	Immunized Animal	Antigen Retrieval	Dilution	Supplier
Collagen 1	Rabbit	Microwave heating in 0.01 mol/L citrate buffer (pH 6.0) at 100 °C, 1 min	1:500	Cell Signaling (Danvers, MA, USA)
ALP	Rabbit	Microwave heating in 0.01 mol/L citrate buffer (pH 6.0) at 100 °C, 1 min	1:200	TAKARA (Kusatsu, Japan)
Osteocalcin	Rabbit	Microwave heating in 0.01 mol/L citrate buffer (pH 6.0) at 100 °C, 1 min	1:1000	Abcam (Cambridge, UK)

ALP—alkaline phosphatase.

## Data Availability

The data presented in this study are available on request from the corresponding author. The data are not publicly available due to ethical restrictions.

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
