# Peer review of "Biological Effects of Bioresorbable Materials in Alveolar Ridge Augmentation: Comparison of Early and Slow Resorbing Osteosynthesis Materials"

_materials, 2021, doi:10.3390/ma14123286_

Round 1
Reviewer 1 Report
The article is very interesting and can be accepted for publication, however Authors should include in their introduction and discussion papers about the following topics:
1) if such material can be useful in systemic patients needing augmentation procedures (please cite PubMed ID31157397)
2) The potential role of such material in combination with stem cells (please cite PubMed ID32811413)
3) The role of digital dentistry in management of such material, in particular when a reconstruction of the aesthetic area is needed (please cite PubMed ID31664999)
After these changes the paper can be acceptable for publication
Author Response
Responses to Reviewers’ Comments
Thank you very much for your invaluable comments and kind acceptance. We have incorporated all the reviewers’ comments and suggestions into our manuscript; the corresponding changes are highlighted in red font in the revised manuscript.
We would like to say thank you once again for the suggestions, which were very helpful in further improving our manuscript.
Comments from Reviewers and Responses
Reviewer 1
The article is very interesting and can be accepted for publication, however Authors should include in their introduction and discussion papers about the following topics:
Comment 1) Reviewer1: 1) if such material can be useful in systemic patients needing augmentation procedures (please cite PubMed ID31157397)
Response:
We thank you for this helpful comment.
"Point-of-care testing for hepatitis C virus infection at an Italian dental clinic: portrait of the pilot study population" is a paper on hepatitis C screening in a dental clinic reported by Simona Tecco and Paolo Capparè. This treatise is very nice and important. Our research needs to understand and apply this wonderful treatise in the future. Unfortunately, we could not focus on the important points of our research. It will be our further challenge.
Comment 2) Reviewer1: 2) The potential role of such material in combination with stem cells (please cite PubMed ID32811413)
Response:
We thank you for this helpful comment.
We added the usefulness of HA by citing a wonderful review article by Paolo Capparè in dissucussion.
Comment 3) Reviewer1: The role of digital dentistry in management of such material, in particular when a reconstruction of the aesthetic area is needed (please cite PubMed ID31664999)
Response:
Thank you for this helpful comment.
Citing excellent literature in the area of aesthetics reported by Francesca Cattoni and Paolo Capparè et al., We added to the discussion section that it is useful as a material for anterior tooth bone augmentation.

Reviewer 2 Report
The article presents an interesting study. The study design is proper, the aim is clearly stated. Conclusions are justified by results. However, I have some suggestions.
Introduction
Please add information what are clinical inductions to use composite screws.
M&M
The control group is missing.
Results
Could you add clinical photos / X-rays.
Discussion
Please compare your results with other studies evaluating screw made of different materials.
What are study limitations?
References
Could you add other references on that topic? Most of the cited literature is published by the authors of this article. Are there other researches investigating this subject?
Author Response
Responses to Reviewers’ Comments
Thank you very much for your invaluable comments and kind acceptance. We have incorporated all the reviewers’ comments and suggestions into our manuscript; the corresponding changes are highlighted in red font in the revised manuscript.
We would like to say thank you once again for the suggestions, which were very helpful in further improving our manuscript.
Reviewer 2
The article presents an interesting study. The study design is proper, the aim is clearly stated. Conclusions are justified by results. However, I have some suggestions.
Comment 1) Reviewer2: Please add information what are clinical inductions to use composite screws.
Response:
Thank you for this helpful comment.
Composed of u-HA and PLLA, this bioabsorbable device is manufactured by a com-pression molding strengthening process and a forging process that incorporates machining. Owing to its composition and special manufacturing process, this device has achieved higher mechanical strength and biological activity.
The literature of clinical use studies using this screw in the maxillofacial region was cited and added in introduction session.
Comment 2) Reviewer2: The control group is missing.
Response:
Thank you for this helpful comment.
Our study compared u-HA / PLLA / PGA with u-HA / PLLA. This study was conducted on humans, and it was difficult to set a control group. I would appreciate your understanding.
Comment 3) Reviewer2: Could you add clinical photos / X-rays.
Response: Thank you for this helpful comment.
We added the figures of CT axial images (Figure 1). Please check it in our manuscript.
Comment 4) Reviewer2: Please compare your results with other studies evaluating screw made of different materials.
Response: Thank you for this helpful comment.
In the discussion section, we added a comparison with similar studies in the past as follows:
Since 2004, Chacon et al.[23] reported block bone grafting with resorbable screws in animal models, a few research papers have been reported. In 2006, Raghoebar et al.[24] reported a comparative study of biodegradable screws consisting of poly-DL-lactide acid (PDLLA) and titanium screws for the fixation of bone grafts in humans. The study reported no histological signs of a significant inflammatory response to poly-DL-lactide acid (PDLLA) material, except for giant cell abundance compared to titanium screws. They also show that fibrous connective tissue intervened between the screw and the bone in PDLLA case. Quereshy et al.[25] published a study using absorptive screws to fix human block grafts, showing that PGA / PLLA screws do not adversely affect graft integration and viability. Unfortunately, this study did not provide a histological evaluation. An important study report showing that our findings compared to previous studies are very good at the histological evaluation of u-HA / PLLA and u-HA / PLLA / PGA in humans.
Comment 5) Reviewer2: What are study limitations?
Response: Thank you for this helpful comment.
In the discussion section, we added limitation to our study as follows; Our research has two limitations. The first limitation is that the number of samples is small. It is difficult to collect many cases at this stage because it targets humans. In the future, it is hoped that the target research period will be extended and that studies will be conducted in many cases. Regarding the second limitation of this study, it was not possible to assess tissue changes associated with these materials over time. In this study, we evaluated the timing of dental implant placement after bone formation and the samples were collected. In particular, u-HA/PLLA/PGA takes a short time to absorb; therefore, it is desirable to evaluate changes over time in the future. However, to the best of our knowledge, the results of this study elucidate the bone healing properties and histological environment of bone healing of u-HA/PLLA and u-HA/PLLA/PGA bioabsorbable materials in human maxillofacial bone; this is the first study to do so. This is an interesting and important description of the in vivo reaction of u-HA/PLLA and u-HA/PLLA/PGA bioabsorbable materials.
Comment 6) Reviewer2: Could you add other references on that topic? Most of the cited literature is published by the authors of this article. Are there other researches investigating this subject?
Response: Thank you for this helpful comment.
We have added other references to this manuscript.

Round 2
Reviewer 2 Report
The manuscript was significantly improved.
I am wondering if following limitations: only 2 screws evaluated, no control group, relatively short follow-up period may apply to you studies.
Author Response
Responses to Reviewers’ Comments
Thank you very much for your invaluable comments and kind acceptance. We have incorporated all the reviewers’ comments and suggestions into our manuscript; the corresponding changes are highlighted in red font in the revised manuscript.
We would like to say thank you once again for the suggestions, which were very helpful in further improving our manuscript.
Reviewer 2
The manuscript was significantly improved.
Comment 1) Reviewer2: I am wondering if following limitations: only 2 screws evaluated, no control group, relatively short follow-up period may apply to you studies.
Response:
Thank you for this helpful comment.
We have added the following description to the limitation of consideration; Our research has two limitations. The first is that the sample size is small and there is no control group setting. It was difficult to enroll many cases and set up control groups in this human-based study. In the future, we hope to study more patients over an ex-tended research duration. Regarding the second limitation of this study, the tissue changes associated with these materials were not assessed over time and are a short follow-up study. In this study, we evaluated the timing of dental implant placement after bone formation and the samples were collected. In particular, u-HA/PLLA takes a long time to absorb; therefore, it is desirable to evaluate changes over time in the future. However, this research method has already been established and evaluated, and a comparison between u-HA/PLLA and u-HA/PLLA/PGA was made on that basis. [8][17] As a result, to the best of our knowledge, the results of this study elucidate the bone healing properties and histological environment of bone healing of u-HA/PLLA and u-HA/PLLA/PGA bioabsorbable materials in human maxillofacial bone; this is the first study to do so. This is an interesting and important description of the in vivo reaction of u-HA/PLLA and u-HA/PLLA/PGA bioabsorbable materials.
Please check it in our manuscript.
